# Identifying latent subgroups of children with developmental delay using Bayesian sequential updating and Dirichlet process mixture modelling

**Patricia Gilholm**[1,2]*, **Kerrie Mengersen**[1,2], **Helen Thompson**[1]

**1** School of Mathematical Sciences, Queensland University of Technology, Brisbane, Queensland, Australia, **2** Australian Research Council Centre of Excellence for Mathematical and Statistical Frontiers, Queensland University of Technology, Brisbane, Queensland, Australia

* p.gilholm@qut.edu.au

## Abstract

Identifying children who are at-risk for developmental delay, so that these children can have access to interventions as early as possible, is an important and challenging problem in developmental research. This research aimed to identify latent subgroups of children with developmental delay, by modelling and clustering developmental milestones. The main objectives were to (a) create a developmental profile for each child by modelling milestone achievements, from birth to three years of age, across multiple domains of development, and (b) cluster the profiles to identify groups of children who show similar deviations from typical development. The ensemble methodology used in this research consisted of three components: (1) Bayesian sequential updating was used to model the achievement of milestones, which allows for updated predictions of development to be made in real time; (2) a measure was created that indicated how far away each child deviated from typical development for each functional domain, by calculating the area between each child's obtained sequence of posterior means and a sequence of posterior means representing typical development; and (3) Dirichlet process mixture modelling was used to cluster the obtained areas. The data used were 348 binary developmental milestone measurements, collected from birth to three years of age, from a small community sample of young children ($N = 79$). The model identified nine latent groups of children with similar features, ranging from no delays in all functional domains, to large delays in all domains. The performance of the Dirichlet process mixture model was validated with two simulation studies.

## Introduction

This research used a three-step ensemble method which incorporated Bayesian sequential updating and Dirichlet process mixture modelling (DPMM) to identify latent subgroups of children who have a similar developmental trajectory, from birth to three years of age, in order

**Data Availability Statement:** The data used in this study contains milestone measurements of infants and young children aged between 1 month and 3 years. This data has been provided and is owned

by The Developing Foundation. Therefore, the authors of this study cannot legally distribute this data. However, the Data and Research Manager at The Developing Foundation, Hugh McKenzie, has agreed to make available the dataset that can be used to reproduce the results in this paper, upon request. To gain access to this data, interested researchers can contact Hugh (hugh@developingfoundation.org.au). The authors confirm that they had no special access privileges to the data that other researchers would not have.

**Funding:** \Work by PG was supported by an Australian Technology Network of Universities Industry Doctoral Training Centre scholarship, co-funded by QUT and the Developing Foundation. The Developing Foundation played a role in data collection. The funders had no role in study design, analysis, decision to publish or the preparation of the manuscript.

**Competing interests:** The authors have declared that no competing interests exist.

to uncover subgroups of children who are experiencing delays in development during these early years. The novelty in this approach is the use of Bayesian sequential updating for modelling the achievement of developmental milestones, which allows for updated predictions to be made at the same time as the child develops. The use of Dirichlet process mixture modelling as the clustering method is also a new approach for this application, which is a more flexible and adaptive clustering approach compared to the common clustering approaches used in developmental research.

The early identification of children who have a developmental disability or delay can sometimes be challenging, as developmental delays may occur gradually and only become more evident as a child grows older [1]. As a consequence, children are often referred to intervention services when they are older than three years of age, which may not coincide with the timing of the delay [2]. An earlier diagnosis may lead to more prompt access to early intervention. Therefore, understanding the development of at-risk children prior to three years of age is necessary in order to facilitate diagnosis and access to early intervention [3].

A common approach that is used to identify at-risk children, during these early years, is to screen and monitor developmental milestones. Developmental milestones are behaviours that are displayed by children at certain times during their development, from infancy through to school age. Monitoring developmental milestones can provide a systematic approach in which to observe the progress of development over time [4, 5]. Developmental milestones have been used in research to classify children into subgroups that describe their developmental functioning, by using unsupervised clustering methods [6–8]. Unsupervised clustering refers to a collection of statistical and machine learning methods that divide cohorts into subgroups based on the structure within the data, when there are no class labels available for classification [9]. Common unsupervised clustering methods include *K*-means [10] and finite mixture modelling [11], which is also known as latent class analysis or growth mixture modelling for longitudinal data [12].

Unsupervised clustering methods have been applied in retrospective studies to identify subgroups of specific developmental disabilities including Attention-Deficit/Hyperactivity Disorder [13], Autism Spectrum Disorder(ASD) [14, 15] and Pervasive Developmental Disorders [16]. Prospective designs have also been used to cluster at-risk infants [17]. However, these studies often only consider a single developmental disorder, such as ASD [18–20], or focus on only one domain of development, such as language development [21, 22] or communication skills [23]. It has been shown that there are many overlapping features among different neurodevelopmental disorders [24], therefore important comorbidities among the disorders can be missed when studying each disorder in isolation [1]. In order to investigate the similarities among the many neurodevelopmental disorders during the early years of development, a diverse community sample of young children was used, which included both typically developing children and children with a variety of developmental disorders and delays, such as Cerebral Palsy and ASD. In addition, to construct a more comprehensive picture of development during these early years, the data used in this research incorporates milestones collected at 28 measurement occasions from birth to three years of age, and includes measurements from six domains of functioning.

The purpose of this research is to implement a more personalised approach to modelling developmental milestones, by first, learning and updating each child's developmental profile as the milestones are met over time, second, comparing each child's developmental profile to that of a typically developing child and, third, identifying latent subgroups of children with similar developmental profiles. The first step of the proposed method uses Bayesian sequential updating to model the probability of milestone achievement. Bayesian sequential updating provides a prediction of behaviour based on the information obtained at previous trials or

measurement occasions. This is achieved by incorporating previous information into the prior, so that past behaviours have some influence on the posterior estimates [25]. This makes it an ideal method for sequentially analysing data that are collected over time, as the likelihood needs to only be calculated for the new data in order to update the model parameters [26]. Bayesian sequential updating is commonly applied to clinical trials, including the continual reassessment method for Phase I clinical trials [27, 28] and Bayesian adaptive design for therapy development [29, 30]. However, to the authors' knowledge, this approach has yet to be applied to modelling developmental milestones.

In the second step, the proposed method summarises the sequence of posterior probabilities obtained from each child by calculating the area between the child's sequence and a reference sequence representing a theoretical child who had achieved all milestones. Inspired by the comparison of Kaplan-Meier survival curves, proposed by Chen et al [31], the rescaled area between the sequences provides a metric that indicates how dissimilar each child is from typical development. The rescaled areas range from 0 to 1, with values closer to 1 indicating larger differences between the sequences [31]. The construction of the areas also aids clustering, as it significantly reduces the dimensionality of the data.

In the third step, Dirichlet process mixture modelling is used as the clustering method. DPMMs have been applied to numerous clustering problems in health, including stratification of children's health [32], classification of Parkinson's disease [33, 34] and classification of fetal heart rates [35]. The DPMM is a Bayesian nonparametric model that introduces uncertainty into the number of clusters through partitioning the data stochastically at each iteration of a Markov chain Monte Carlo (MCMC) sampler [32]. This approach has a distinct advantage over traditional clustering methods, such as finite mixture modelling and $K$-means, as it allows the number of clusters to be dictated by the data, meaning that the analyst does not need to specify the number of clusters *a priori* [32]. This flexibility is important for the current application, as the data will increase as children respond to more milestones, or more children join the program. By using a DPMM, the number of clusters can also increase or merge as new data are collected and included in the model.

Through using this three-step approach to model the achievement of developmental milestones, this research aimed to identify subgroups of children who were experiencing similar developmental delays across six functional domains. By applying this modelling approach, individual predictions of development can be made and updated for each functional domain and the obtained subgroups can be used to assist treatment planning by targeting the specific developmental delays that are characteristic of each subgroup.

## Materials and methods

### Data

The data for this study were provided by *The Developing Foundation* [36], a Brisbane-based Australian charity that supports families who are seeking treatment for a family member with a brain injury or developmental disability. The organisation collected data on developmental milestones using an online program, *Developing Childhood* [37]. The program allows parents and carers to assess and track their child's achievement of developmental milestones from birth to three years of age. There are 348 milestones in total, which are categorised into six functional domains: Vision, Auditory, Tactile, Movement, Speech and Hand function. Fifty-eight milestones are measured within each of these functional domains. The milestones are not measured uniformly across time; within each functional domain, there are three ordered milestones measured per month in the first 12 months, two ordered milestones measured per month between 13 to 18 months and one milestone measured per month from 19 to 25

**Table 1. Example 1, 12, 18 and 34 month milestones in each functional domain.**

| Functional Domain | 1 month | 12 months | 18 months | 34 months |
|---|---|---|---|---|
| Vision | Instantly blinks at bright light | Television or colourful moving objects capture attention | Visually aware of close and distant world | Recognises and points out tiny details in pictures |
| Auditory | Instantly startles to sudden loud noise | Listens to speech without distraction from other sounds | Follows simple two-step commands | Comprehends three key words in a sentence |
| Tactile | Negative response to pain, positive to comfort | Maintains balance with supported stepping | Begins to identify objects by touch alone | Aware of body size in relation to surroundings |
| Speech | Non-specific cry | Sound-making with intent | Social speech used for interacting | Regular use of speech to tell stories and experiences |
| Movement | Unrestricted range of movement in all limbs | Walks holding on to one hand | Attempts to run but without a lot of control | Can pedal a tricycle with good control |
| Hands | Hands mostly fisted or slightly open | Finger feeding with pincer grasp | Stacks 4-6 blocks | Can dress and undress completely |

months. The remaining three milestones are measured at 28, 31 and 34 months. The order of the milestones was determined by developmental experts at *The Developing Foundation*. Example milestones for each functional domain are shown in Table 1.

## Participants

The original sample consisted of data from 118 children whose parents or carers were voluntarily using the program. This sample consists of both typically developing children and children with a diverse range of developmental disabilities, including Autism Spectrum Disorder, Cerebral Palsy, Down Syndrome, and speech and hearing impairments, as well as more general developmental delays. Although the nature of each child's developmental status is confidential, it is assumed that this sample consists of a larger proportion of children with a developmental disorder or disability than in the general population, as the program was specifically designed for families who seek assistance from *The Developing Foundation*. The QUT University Human Research Ethics committee waived the need for consent from the parents or guardians for the data used in this research, as the data does not contain any identifiable information. In order to develop the method, only children with complete data sequences were included in the analysis. Extensions to accommodate missing data are described in the Discussion. Of the original sample, complete data sequences were available for 79 children.

## Method

**Bayesian sequential updating.** A child's achievement of the milestones is represented as a sequence of Bernoulli trials. The milestones are assumed to be independent, where milestone achievement is recorded as $y = 1$ and not achieving a milestone is recorded as $y = 0$. This is considered a reasonable assumption as milestone achievement is not necessarily cumulative, in that some children can achieve later milestones without achieving earlier ones. Moreover, the dependency between milestones achieved for each child is modelled through the sequential updating of the prior. However, in order to investigate the independence assumption, a sensitivity analysis was performed, and the results indicated that this assumption is reasonable. This analysis is addressed in more detail in the Sensitivity analyses section of the Results.

Bayesian sequential updating is a recursive process that can be used for trials that are observed in a sequence, whereby the posterior distribution for the observation(s) in the first trial becomes the prior distribution for the observation(s) in the second trial. The sequential

updating of the prior distribution for a series of Bernoulli trials is a simple procedure, as the posterior distribution for $z$ successes out of $N$ trials using a $Beta(\theta|a, b)$ prior has a posterior distribution of the form $Beta(\theta|z + a, N - z + b)$ [38]. Therefore, for sequential data, the posterior distribution can be updated for each new observation by adding 1 to $a$ for each subsequent success or 1 to $b$ for each subsequent failure. A brief summary of the Bayesian beta-Bernoulli model is provided in Appendix A of S1 Appendix.

To perform the sequential updating, a $Beta(1, 1)$ prior is used for the first observation for all participants, as this is a uniform prior with equal probability of success or failure in achieving a milestone. The sequential updating procedure is then implemented for each individual child, resulting in a series of posterior means, which represents the probability of achieving each milestone based on the child's past milestone achievements.

To visualise the probability of milestone achievement for each child over time, we plotted the posterior means for the observed milestones across time for each functional domain, along with their 95% highest posterior density (HPD) intervals. A selection of plots of the posterior means for six children in the Auditory functional domain are displayed in Fig 1.

**Area between posterior probability sequences.** In order to compare and cluster the sequences of posterior means, a child's sequence of posterior means is compared to a theoretical typically developing child's sequence, by calculating the area between the sequences. The theoretical "gold-standard" sequence of posterior means is created by performing Bayesian sequential updating on simulated data for a hypothetical child who achieves all milestones.

As the sequences of posterior means are stepwise functions, the area between the sequences can be calculated as follows

$$area(F_1, F_2) = \frac{1}{w}\left\{|F_2(t_n) - F_1(t_n)|(w - t_n) + \sum_{i=1}^{n}|F_2(t_i) - F_1(t_i)|(t_{i+1} - t_i)\right\}, \tag{1}$$

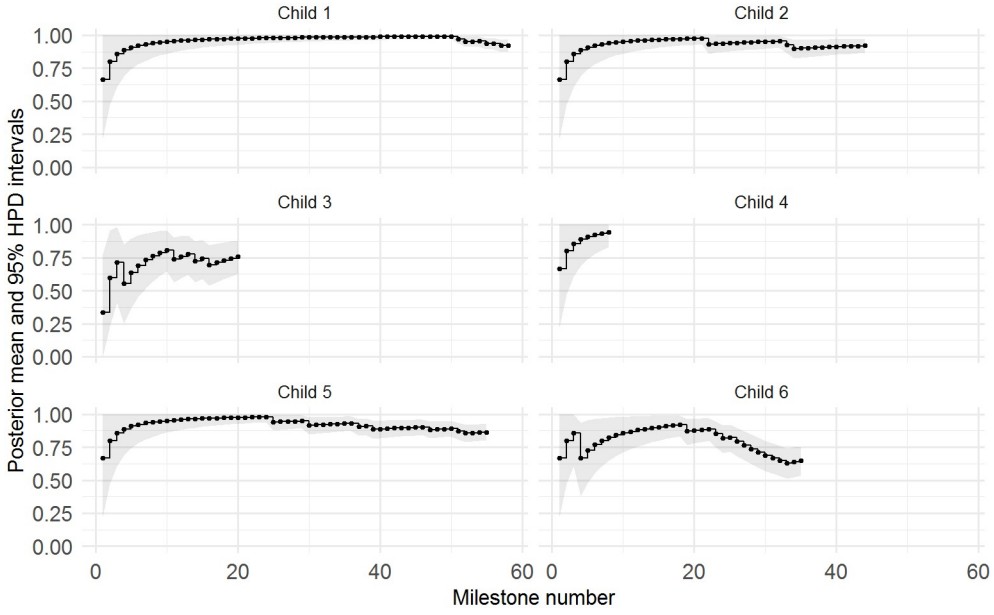

**Fig 1. Posterior means and 95% HPD intervals for the milestones in the Auditory functional domain for six children.** This figure shows variability among the children in terms of the number of milestones recorded for each child, as well as the progress of development over time. For example, Child 1 responded to all of the milestones in the Auditory functional domain, has very high posterior means for most milestones and only starts to show a slight decline at around the 50th milestone. In contrast, the posterior means for Child 6 are much more variable, with a steeper decline beginning at the 20th milestone.

where $F_1$ is the step function of the child's posterior means, $F_2$ is the "gold-standard" development step function, $t$ denotes the discrete milestone time points ranging from $t = 1 \leq i \leq n$ and $w$ corresponds to the number of observed milestones [31]. As the number of observed milestones varies across children, the areas are rescaled by the total number of milestones observed by each child, $w$. This results in a rescaled area between 0 and 1, where scores closer to 0 indicate children whose posterior means are more similar to the "gold-standard" posterior means. An example of the area that is calculated is displayed in Fig 2.

In this application, all 79 children started their milestone measurements in month 1, but this may not always be the case. If children do not begin their milestone measurements in the first month (i.e., there are measurements missing before the beginning of the sequence), the starting point for the reference sequence can be set equal to the starting point of the child's sequence, in order for the reference sequence to remain the same for all children. Six areas are calculated for each child, one for each functional domain. In this application, the resulting areas were highly positively skewed with many scores close to 0. In order to assist clustering, the areas were transformed from the [0, 1] scale to $(-\infty, +\infty)$ using the logit transformation.

**Dirichlet process mixture model.** The Dirichlet process mixture model is a Bayesian nonparametric method for unsupervised clustering. A general description of the DPMM is available in Appendix B of S1 Appendix. The *stick-breaking representation* for drawing samples from a Dirichlet process was used, which was first established by Sethuraman [39]. In this representation, the mixing distribution $G$ is represented by an infinite sum of weighted point masses:

$$G = \sum_{k=1}^{\infty} C_k \delta_{\theta_k}, \tag{2}$$

where $\delta_{\theta_k}$ represents a point mass of 1 located at $\theta_k$ which is sampled directly from the base distribution, $G_0$, i.e., $\theta_k \sim G_0$ [40]. The weights $C_k$ are generated sequentially through the stick-

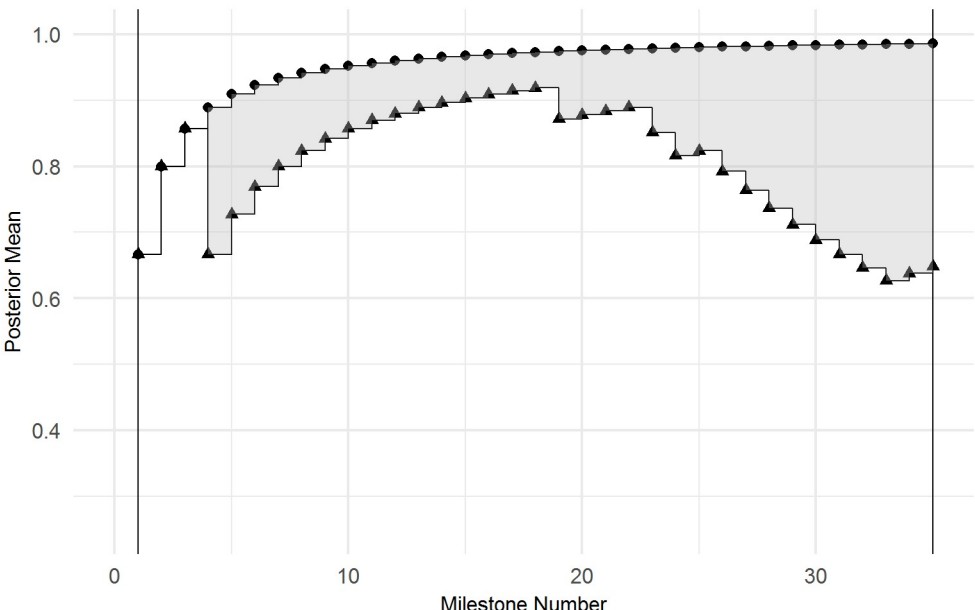

**Fig 2. Example area calculation.** The shading represents the absolute area calculated between the theoretical "gold-standard" posterior means (circles) and the child's posterior means (triangles), which is then rescaled by the number of observed milestones. Note that the posterior means are the same for the first three milestones, resulting in an area of 0 for these observations.

breaking process:

$$V_1, V_2, \ldots \overset{iid}{\sim} Beta(1, \alpha)$$
$$C_1 = V_1$$
$$C_k = V_k \prod_{j=1}^{k-1}(1 - V_j); \qquad k \geq 2. \tag{3}$$

The stick-breaking analogy refers to the generation of the weights, where the stick starts with a length of one and the first weight is broken off from the stick at length $C_1$. The remaining stick has a length of $1 - C_1$ and $C_2$ is broken off from this length of stick [41]. This process continues for each successive break, where the stick can theoretically be broken an infinite amount of times.

Posterior inference from a DPMM utilises Markov chain Monte Carlo (MCMC) posterior simulation [42]. A number of different methods have been established that use Gibbs sampling, including blocked sampling [43], retrospective sampling [44] and slice sampling [40]. This research implemented the slice sampling procedure, established by Walker [40]. An outline of the slice sampler is provided in Appendix C of S1 Appendix.

Due to the nature of the stick-breaking construction of the Dirichlet process, there is a size-biased ordering of the expected prior mixture probabilities, e.g., $E[C_k] > E[C_{k+1}]$ for all $k$ [45]. Therefore, the Gibbs sampler needs to adequately mix over the cluster labels, otherwise clusters with lower labels will be given higher prior probability [46]. In order to prevent the Gibbs sampler from getting stuck in local modes corresponding to one assignment of cluster labels, label-switching moves were implemented as outlined by Papaspiliopoulos et al [44].

The Dirichlet process mixture model was implemented, as outlined above, to model a mixture of $p$-dimensional multivariate normal distributions, whereby, conditional on each cluster $k$, the likelihood for $y_i$ is

$$p(y_i|z_i = k, \boldsymbol{\mu}_k, \Sigma_k) = MVN(\boldsymbol{\mu}_k, \Sigma_k) \tag{4}$$

with mean $\boldsymbol{\mu}_k = [\mu_{1k}, \ldots, \mu_{pk}]$ and variance-covariance matrix $\Sigma_k$. A joint prior distribution $p(\boldsymbol{\mu}_k, \Sigma_k) = p(\boldsymbol{\mu}_k|\Sigma_k)p(\Sigma_k)$ was used, similar to van Havre et al [47], where

$$p(\boldsymbol{\mu}_k|\Sigma_k) = MVN(\boldsymbol{b}_0, \Sigma_k/N_0)$$
$$p(\Sigma_k) = IW(c_0, C_0). \tag{5}$$

The prior distribution for the concentration parameter, $\alpha$ (in Eq 3), was $Gamma(\eta_1, \eta_2)$, which is commonly used for DPMMs [48].

As each iteration of the MCMC Gibbs sampler estimates the number of clusters, post-processing methods are required to obtain the optimal number of clusters over all iterations. The partitioning around medoids (PAM) method [49] was used as the post-processing method, which is an algorithm that searches for $k$ representative objects, or medoids, and then forms clusters by assigning each remaining object to the nearest medoid [49]. Due to the label switching moves, it is not possible to simply assign the most frequent cluster label, across the iterations, for each observation. Alternatively, the posterior similarity matrix, $S = P(z_i = z_j|y)$, was calculated, which is an $n \times n$ matrix containing the pairwise proportion of iterations that two observations were assigned to the same cluster [50]. The dissimilarity matrix, $1 - S$, was then used as input for the PAM algorithm. The algorithm was run for $k = 2$ to $k = 20$ medoids and the different clusterings were compared using the average silhouette width, which describes how well each object fits to their assigned cluster [51].

## Results

### Dirichlet process mixture model

The Bayesian sequential updating and area calculations were performed for the six functional areas for each child. These areas were then used as input in the Dirichlet process mixture model. R code for performing the Bayesian sequential updating and area calculation, as well as an example of using this code on simulated data is available on Github [52].

Before running the model, a grid experiment was performed to observe the effect of different hyperparameter specifications on the number of clusters obtained from the model. The selection of hyperparameters chosen for the grid experiment were guided by the literature, where similar hyperparameters have been used [48, 53–57]. The details and results of this grid experiment can be found in S2 Appendix. Based on these results, the optimal hyperparameters for the prior distributions, outlined in Eq 5, were $\mathbf{b}_0 = \bar{\mathbf{y}}$, $N_0 = 0.1$, $c_0 = 7$, $C_0 = \Sigma_y$. In addition, the prior distribution for $\alpha$ was a $Gamma(1, 1)$ distribution. Three chains of the DPMM slice sampler were specified for 1,000,000 iterations. The three chains were initialised using $K$-means, with the number of clusters defined as $K = 5$, $K = 10$ and $K = 15$, respectively. R statistical software [58] was used to conduct the slice sampling. The slice sampling code is publicly available on Github [59].

Convergence was achieved for this model based on a Gelman-Rubin statistic of less than 1.1 for both $K$ and $\alpha$ ($GR_K = 1.01$, $GR_\alpha = 1$). Once the model had converged, the optimal clustering was determined by calculating the average posterior similarity matrix across the three chains and finding the optimal partition by using the PAM algorithm. The number of clusters specified for the PAM algorithm ranged from 2 to 20, and the different clusterings were compared by calculating the average silhouette width (see S2 Appendix for details). Through this process, 9 clusters were found to be optimal.

The sample means and standard deviations for the areas of each group, as well as the group sizes can be found in Table 2 and the profiles for each group can be found in Fig 3. The main characteristics of the nine groups are as follows. Group 1 is the largest group, consisting of 22 children with relatively small areas for each functional domain. This group contains a number of outlying individuals, whose profiles do not fit with the characteristics of the other groups. Group 2 contains eight children who have large areas for all functional domains. Group 3 contains nine children and is characterised by larger areas for the auditory domain, some non-typical development for the speech, tactile and vision domains and close to typical development for hand function and movement. Group 4 only contains three children, who have larger areas for the hand function and movement domains. Group 5 consists of nine children and is characterised by some deficits in the auditory, tactile and vision domains, and typical development

**Table 2. Group size, mean area and (standard deviation) for each functional domain, per cluster.**

| Group | Group size | Auditory | Hands | Movement | Speech | Tactile | Vision |
|---|---|---|---|---|---|---|---|
| 1 | 22 | 0.040 (0.074) | 0.012 (0.035) | 0.024 (0.059) | 0.052 (0.104) | 0.024 (0.075) | 0.001(0.001) |
| 2 | 8 | 0.149 (0.115) | 0.142 (0.153) | 0.085 (0.063) | 0.157 (0.171) | 0.077 (0.060) | 0.132(0.091) |
| 3 | 9 | 0.123 (0.082) | 0.034 (0.053) | <0.001(<0.001) | 0.074 (0.122) | 0.030 (0.030) | 0.087(0.122) |
| 4 | 3 | 0.010 (0.008) | 0.081 (0.023) | 0.149 (0.118) | 0.003 (0.004) | <0.001 (0.000) | 0.004(0.006) |
| 5 | 9 | 0.074 (0.065) | 0.008 (0.016) | 0.004 (0.006) | 0.002 (0.004) | 0.063 (0.060) | 0.054(0.047) |
| 6 | 8 | 0.074 (0.071) | 0.001 (0.001) | 0.068 (0.059) | 0.167 (0.139) | 0.065 (0.063) | 0.060(0.061) |
| 7 | 7 | 0.001(<0.001) | 0.002 (0.003) | 0.001 (0.001) | 0.014 (0.017) | 0.015 (0.024) | 0.016(0.013) |
| 8 | 10 | 0.001 (0.001) | 0.001(<0.001) | 0.001 (0.001) | 0.001(<0.001) | <0.001(<0.001) | 0.001(0.001) |
| 9 | 3 | 0.043 (0.033) | <0.001 (0.000) | 0.022 (0.055) | 0.297 (0.161) | <0.001 (0.000) | 0.083(0.056) |

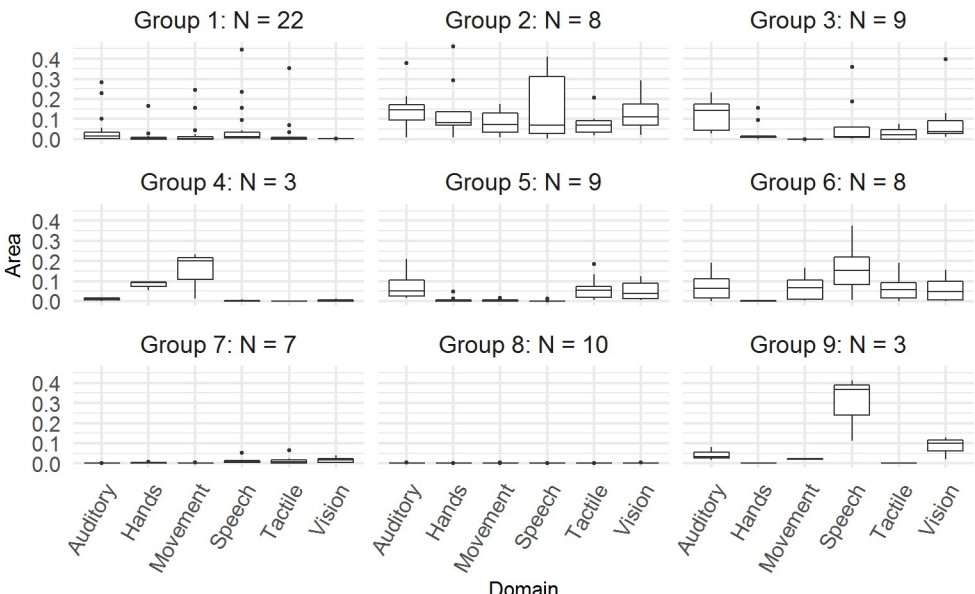

**Fig 3. Cluster profiles and sample size of the nine subgroups.** Each panel corresponds to a subgroup's profile. Boxplots of the areas between the posterior means for each functional domain are represented within each panel. Higher scores indicate larger differences between the group's posterior means and the posterior means representing typical development.

for the remaining domains. Group 6 contains eight children who have larger areas in all functional domains except for hand function. Group 7 consists of seven children who have achieved most milestones, across all domains, and they differ from Group 8, containing ten children, who have achieved all milestones. Finally, Group 9 consists of three individuals who have very large speech deficits. Additional plots that display the cumulative sum of the achieved milestones for each group can be found in S1 Fig.

Groups 4 and 9 have the smallest cluster sizes and therefore could be considered outliers. Alternatively, these groups could represent emerging clusters that would have a larger representation if more data were collected. Similarly, the outlying observations in Group 1 may also split to form smaller, representative clusters when additional information is collected from new observations. The uncertainty in the number of clusters is a key feature of DPMMs and allows for more nuanced groupings to emerge from the data. This is particularly important in the context of child development, as even small deviations from typical development can have an impact on future functioning [60, 61].

Additional analyses using two alternative clustering algorithms, namely, k-means and model-based clustering were undertaken to compare the performance of the DPMM to these commonly used clustering methods. In summary, K-means was only able to cluster the data into two groups, representing typical and atypical development. Model-based clustering performed slightly better as it was able to identify two smaller clusters, in addition to the typical/ atypical clusters. These smaller clusters were equivalent to clusters 5 and 8 from the DPMM. Overall, the DPMM was able to identify smaller, more distinct clusters than these alternative methods, which is important for the current application where identifying smaller groups for targeted intervention is the goal. A detailed overview of these methods and the results of these additional analyses can be found in S3 Appendix.

In addition, we compared the PAM post-processing method to the least-squares clustering approach [62]. Briefly, this method selects a clustering which minimizes the sum of squared

deviations from the posterior similarity matrix. This method returned many more clusters than the PAM method, with an average of 16 clusters selected. This was the case when examining each chain separately, as well as combined. Sixteen clusters was considered excessive for the current application, and many of these clusters had a sample size of $N = 1$. Therefore, the clusters obtained from this method were not explored further.

## Sensitivity analyses

Two sensitivity analyses were performed to assess different aspects of the model. In the first sensitivity analysis, two simulation studies were performed to illustrate how the proposed DPMM performs for (a) well separated, adjacent or overlapping clusters (scenario 1) and (b) small, medium or large sample sizes (scenario 2). The second sensitivity analysis assessed the independence assumption of the milestones and is briefly described at the end of this section.

In the first sensitivity analysis, for each scenario, three bivariate clusters were simulated using the clusterlab package in R [63]. For scenario 1, three small clusters, with 50 observations in each cluster, were simulated and compared under three conditions to assess the DPMM's performance when clusters are overlapping. A small sample size was chosen in order to make comparisons to the application data, which also has a small sample size. In the first condition, the three clusters were visibly well-separated, in the second condition the three clusters were adjacent, but not overlapping, and in the third condition, the three clusters were slightly overlapping. The simulated data used in scenario 1 are displayed in the first row of Fig 4. The 15 hyperparameter specifications that were used for the models in the grid experiment were also used in the simulation study (see Table 1 of S4 Appendix). Three chains were specified for each model and each chain ran for 1,000,000 iterations. The Gelman-Rubin statistic for $K$ and $\alpha$, the average silhouette width for three clusters specified using the PAM method, and the classification accuracy for each model are displayed in Table 2 of S4 Appendix.

For conditions 1 and 2, the DPMM returned exactly the same clusters that were simulated, regardless of which hyperparameters were used. This was not the case for condition 3, where

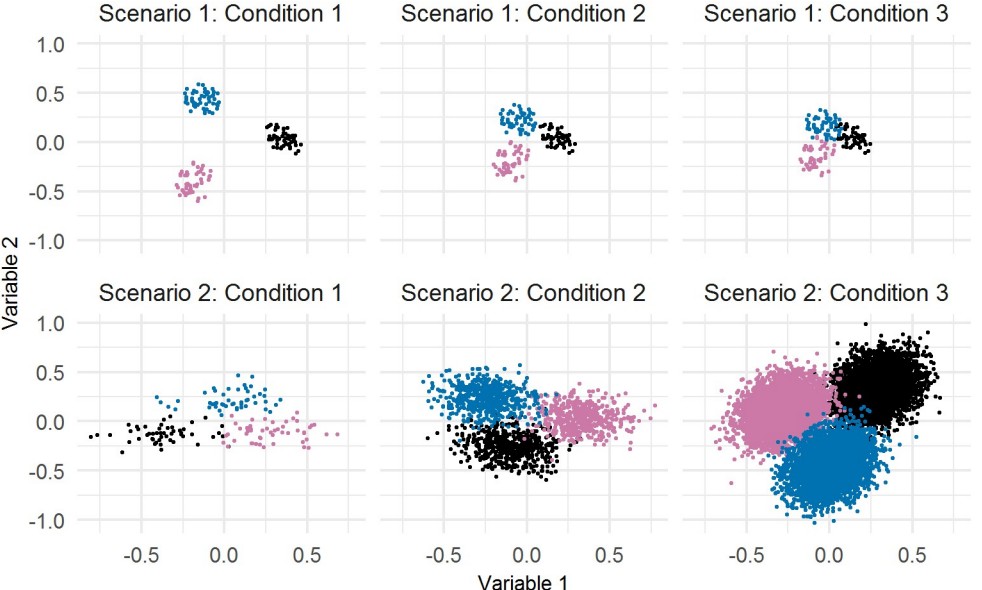

**Fig 4. Simulated data used for each condition in the sensitivity analysis.** The first row contains the data used for scenario 1 and the second row contains the data used for scenario 2.

10 out of the 15 models returned 3 clusters as the optimal number of clusters (based on the average silhouette widths from the PAM method), and the remaining 5 models returned 2 clusters. For the models that returned 3 clusters, the boundary for the clusters varied, resulting in some misclassification of cases (range of classification accuracy = 76.67% to 94.00%). This demonstrates that, even for relatively simple cases, clustering using a DPMM can result in different clustering solutions due to the uncertainty that is introduced into the number of clusters when the data are small, noisy and overlapping. For this example, there was an improvement in the classification accuracy when the precision parameter, $N_0$, ranged from 0.1 to 0.5. Values above or below these cut-offs did not perform as well, as they either returned 2 clusters as the optimal solution or the classification accuracy reduced. The precision parameter has an influence on the dispersion of the cluster means and should be considered carefully for each application in order to identify the optimal dispersion [54].

The second simulation study was undertaken to identify if the difficulties associated with the clustering of overlapping, noisy data would remain if more observations were collected. For this scenario, three slightly overlapping bivariate clusters were sampled, which differed in terms of sample size for each condition. For condition 1, each cluster consisted of 50 observations, for condition 2, each cluster contained 500 observations and for condition 3 there were 5000 observations in each cluster. The sample sizes were selected such that condition 1 was the same size used in scenario 1 and conditions 2 and 3 increased the sample size by a magnitude of 10. The simulated data used for scenario 2 is displayed in the second row of Fig 4. For this simulation, the 15 models ran for 40,000 iterations. The number of iterations were less than those used in the previous applications as the average run time for the large sample size, using only 10,000 iterations, was 16hr:37min to run the slice sampler. In addition, on average, 55.63GB of RAM was required to calculate the mean posterior similarity matrix for each model. All models were run on a HPC cluster, inclusive of Intel E5-2670, E5-2680v2, E5-2680v3 and 6140 CPU models. Due to the time requirements, the large sample size was assessed by running each chain for segments of 10,000 iterations. Each segment (except the first) was initialised using the values from the last iteration of the preceding segment. The chains were assessed for convergence each time a new segment was added. As the number of iterations used for this analysis was substantially smaller than that used for the application or the first simulation study, 6 chains were run for each model (initialised at $K$ = 2, 4, 5, 6, 10 and 15 clusters) in order to be more certain that the models had converged. The same 15 hyperparameter combinations that were used previously were also used here. After the chains had run for 40,000 iterations, 8 of the 15 models had reached convergence for $K$ and all the models had converged for $\alpha$ based on the Gelman-Rubin statistic. In order to accurately compare the small, medium and large conditions, 6 chains of 40,000 iterations were also specified for the small and medium sample sizes. All the models for these conditions converged, except for one model for the medium sample size. Only the converged models were processed using the PAM method.

The average silhouette width when $K$ = 3 and the classification accuracy for each model, within each condition, are displayed in Table 3 of S4 Appendix. All of the converged models for all conditions returned 3 clusters as the optimal number, based on the average silhouette width, and the average classification accuracy across all models for all conditions was high (small = 95.33%, medium = 96.48%, large = 98.59%), despite the time and memory restrictions associated with processing the largest sample size. There were no major differences across models for each scenario, indicating that the prior specification does not influence the clustering when there is only a slight amount of overlap in the clusters, particularly when the sample size increases. The milestone data is much more complex, resulting in more uncertainty in the number of clusters. These simulation results indicate that the clustering accuracy slightly improves with more observations, but with a much larger computational cost.

A second sensitivity analysis was performed to assess the independence assumption of the milestone measurements. The order of the milestones within each month was randomised in order to investigate the impact of rearranging the order of the milestones. The sequential updating and area calculation were performed on the rearranged milestones, and the results compared to the original ordering of the milestones. This procedure was repeated 10 times with new random orders. The results revealed only minor differences between the areas obtained from the original order and the random order (overall mean difference = −0.00047, overall mean standard error = 0.00093). The full results of this sensitivity analysis can be found on the first author's Github [52].

## Discussion

This research used an ensemble method for modelling and clustering developmental milestones which incorporated Bayesian sequential updating and Dirichlet process mixture modelling. Using Bayesian sequential updating, the probability of achieving each milestone was modelled based on each child's own sequence of milestone measurements. This sequence of probabilities was summarised by calculating the area between each child's sequence and a reference sequence representing "gold-standard" development. The areas were then clustered using DPMM to identify subgroups of children who were experiencing similar delays in development.

This detailed method allows for personalised predictions of milestone achievements to be made, as the updated sequences are constructed using only the child's measurements. The model also introduces uncertainty into the predictions, as each probability of milestone achievement is modelled as a posterior distribution of credible values. This means, in practice, that more detailed predictions can be communicated to parents regarding their child's likely trajectory of development and the certainty associated with each prediction can be conveyed. To develop the method, a static data set was used and the method was implemented retrospectively, however, by using Bayesian sequential updating, this method could be implemented prospectively, where predictions could be made as the child develops, as the method allows for predictions to be easily updated with the collection of new data. By clustering the probability sequences, children who are experiencing similar delays are able to be identified, meaning that early interventions can be tailored to meet the needs of each group, allowing for more personalised treatment planning.

By using this approach, in the present application, nine groups were identified that differed in terms of their level of deviation from typical development, across six functional domains. Although some of the cluster sizes were considered small, these groups represented children that did not have the same developmental pattern as the larger groups. Instead of being placed with the most likely group, which is typical for other unsupervised clustering methods (e.g., $K$-means), these children were placed in their own emerging cluster group. This is important for clinical practice, as these children can have treatments tailored to meet the unique characteristics of the emerging cluster, rather than have tailored treatments based on clusters that they are "most alike", which may not adequately address the needs of the child.

Despite the practical advantages of using this modelling approach, there are a number of methodological limitations that need to be taken into consideration. Firstly, the developmental milestones within each month were assumed to be sequential, based on information elicited from a domain expert. However, the milestones may not be met in this exact order for every child. A sensitivity analysis assessing the independence assumption revealed only small differences in the outcome if the milestones were rearranged within each month. Given this small difference, it is reasonable to assume independence for the milestones within each month and

a more general model could use the binomial distribution to model the milestones, but this approach was not considered here.

Another limitation is that this model did not incorporate any covariates. Covariate information was not available for the sample that was used in this study, but there are a number of covariates that could have an influence on milestone achievement. Past studies have found significant environmental and prenatal predictors of developmental delay, including birth complications and maternal education [64], poverty and caregiver cognitive impairment [65], and low birth weight [66]. The method developed in this paper could benefit from incorporating this type of covariate information into the model, to create more accurate predictions. Finally, the method developed in this paper is most effective with complete data sequences, as it can overestimate the degree of delay when there are missing data points. Imputation or functional data approaches could be explored to rectify this problem, but these approaches were outside the scope of this research. Modelling the milestones using a functional data approach will be explored in future work.

Additional considerations need to be made when using DPMM. Despite its advantages over other clustering methods, several modelling decisions need to be made in order to obtain the most efficient results, including hyperparameter choice, method for sampling from the posterior and technique used for post-processing the MCMC chains. Each one of these aspects of modelling using a DPMM needs to be carefully considered, as different choices can have an influence on the clusters.

In this paper, the slice sampler was selected as the method for sampling from the posterior distribution of the Dirichlet process. There are, however, several alternative samplers that can be used, for example, the truncated sampler [43] and the retrospective sampler [44]. The slice sampler was used in this application as it adaptively selects the number of mixture components [67] and easily updates them at each iteration [68]. Also, unlike the truncated methods, it targets the true posterior rather than an approximation [69]. However, this method does have some limitations. Due to the high correlation between each slice from the slice sampler and the mixture weights, the number of components sampled at each iteration can be large if the slice is small [69, 70]. This can result in slow mixing and high autocorrelations [67], as was the case in this research. However, as these samplers are often developed and illustrated on simulated or low-dimensional datasets, it is likely that similar problems would be encountered using alternative samplers when applied to complex data, such as that used in the current application [45].

The final aspect of DPMM that requires consideration is the choice of method for post-processing the MCMC chains to obtain the optimal number of clusters. The PAM method was used in this paper, however, alternative methods have been proposed that also use the posterior similarity matrix, including Binder's loss function [71], the Posterior Expected Adjusted Rand index [50], and hierarchical clustering [72]. The PAM method was chosen as it consistently assigns individuals to clusters based on the structure of the posterior similarity matrix [73], and has been found to perform better than alternative methods, such as $k$-medoids [51]. However, this method can be computationally intensive when applied to large datasets.

## Conclusion

Overall, the DPMM approach presented here allows for flexibility in modelling and does not require the specification of the number of clusters *a priori*. Additionally, the DPMM takes into account emerging clusters, which makes it ideal for the current application, as it is expected that the clusters will grow or merge as more data are collected. When combined with the Bayesian sequential updating and the calculation of the area between the posterior probability

sequences, this ensemble method demonstrates a new approach to modelling developmental milestones, which can provide detailed information regarding a child's development. This will be able to assist in the formulation of personalised early interventions targeted for developmental delays that occur throughout the early, most critical, years of development.

## Supporting information

**S1 Fig. Cumulative sum plots.** Cumulative sum of the achieved milestones for each functional domain for each group.
(PDF)

**S1 Appendix. Additional model descriptions.** Appendix A: A brief summary of the Bayesian beta-Bernoulli model. Appendix B: A brief introduction to Dirichlet Process Mixture models. Appendix C: An overview of slice sampling.
(PDF)

**S2 Appendix. Grid experiment for selecting hyperparameters.**
(PDF)

**S3 Appendix. Comparison of DPMM to K-means and model-based clustering.**
(PDF)

**S4 Appendix. Results of sensitivity analysis.**
(PDF)

## Acknowledgments

The data used in this research was generously provided by The Developing Foundation.

## Author Contributions

**Conceptualization:** Patricia Gilholm, Kerrie Mengersen, Helen Thompson.

**Formal analysis:** Patricia Gilholm.

**Methodology:** Patricia Gilholm.

**Project administration:** Patricia Gilholm.

**Supervision:** Kerrie Mengersen, Helen Thompson.

**Writing – original draft:** Patricia Gilholm.

**Writing – review & editing:** Kerrie Mengersen, Helen Thompson.

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
