## [Decision Letter · Decision Letter 0]

9 Jan 2020

PONE-D-19-29401

Identifying latent subgroups of children with developmental delay using Bayesian sequential updating and Dirichlet process mixture modelling

PLOS ONE

Dear Miss Gilholm,

Thank you for submitting your manuscript to PLOS ONE. After careful consideration, we feel that it has merit but does not fully meet PLOS ONE’s publication criteria as it currently stands. Therefore, we invite you to submit a revised version of the manuscript that addresses the points raised during the review process.

 Please fully address all helpful and useful comments from the reviewers, including assumption of independence between milestone measurements over time (address how the prior affect the likelihood in case of failure in earlier stage and successful in later stage; justify whether the milestones are ordered appropriately and independent and incorporate sensitivity analyses to test the assumption), data missingness at the beginning of the trajectories and its correlation with the groupings, result comparison with Dahl (2006) method using similarity matrix, assessment of model perform under non-informative priors, consideration of alternative sampling schemes to increase the mixing, posterior distributions for the group means for the resulting clusters, hyperparameters description.

In addition, please also address the following issues:

1) Methods:

a) Timing of milestones

This study developed a Bayesian sequential updating modeling which counted the cumulative developmental milestone achievement over the first 3 years of life in 79 young children with different levels of development and disability in Australia. However, timing was not considered and hence earlier detection of deviation from typical development could not be achieved for early intervention.

b) Alternative modeling

The authors should consider assessing the modeling performance by comparing conventional standard methods for generating latent classes with the studied Bayesian sequential updating modeling for adding subsequent milestones and the Dirichlet process mixture modeling for clustering deviated area from typical development.

We would appreciate receiving your revised manuscript by Feb 23 2020 11:59PM. To enhance the reproducibility of your results, we recommend that if applicable you deposit your laboratory protocols in protocols.io, where a protocol can be assigned its own identifier (DOI) such that it can be cited independently in the future. For instructions see: http://journals.plos.org/plosone/s/submission-guidelines#loc-laboratory-protocols

We look forward to receiving your revised manuscript.

Kind regards,

Man Ki Kwok

Academic Editor

PLOS ONE

Journal Requirements:

2. You indicated that you had ethical approval for your study.

In your Methods section, please ensure you have also stated whether you obtained consent from parents or guardians of the minors included in the study or whether the research ethics committee or IRB specifically waived the need for their consent.

'Work by Patricia Gilholm was supported by an Australian Technology Network of Universities Industry Doctoral Training Centre scholarship, co-funded by QUT and the Developing Foundation. Furthermore, the data used in this research was generously provided by The Developing Foundation.'

"Work by PG is partially funded by the Developing Foundation (https://www.developingfoundation.org.au/). The Developing Foundation also generously provided the data for this study."

Please provide an amended Funding Statement that declares *all* the funding or sources of support received during this specific study (whether external or internal to your organization) as detailed online in our guide for authors at http://journals.plos.org/plosone/s/submit-nowPlease state what role the funders took in the study.  If any authors received a salary from any of your funders, please state which authors and which funder. If the funders had no role, please state: "The funders had no role in study design, data collection and analysis, decision to publish, or preparation of the manuscript."

Reviewers' comments:

Reviewer's Responses to Questions

**Comments to the Author**

1. Is the manuscript technically sound, and do the data support the conclusions?

Reviewer #1: Yes

Reviewer #2: Yes

2. Has the statistical analysis been performed appropriately and rigorously? 

Reviewer #1: Yes

Reviewer #2: Yes

3. Have the authors made all data underlying the findings in their manuscript fully available?

Reviewer #1: No

Reviewer #2: Yes

4. Is the manuscript presented in an intelligible fashion and written in standard English?

Reviewer #1: Yes

Reviewer #2: Yes

5. Review Comments to the Author

Reviewer #1: Review: Identifying latent subgroups of children with developmental delay using Bayesian sequential updating and Dirichlet process mixture modelling

December 26, 2019

Overall, the paper is very well written. The analysis is rigorous and does a great job of ensuring that the model performance is reasonable for inference.

1. Is the assumption of independence between the same milestone measurements over time reasonable? The author’s state “This is considered a reasonable assumption as milestone achievement is not necessarily cumulative, in that some children can achieve later milestones without achieving earlier ones. Moreover, the dependency between milestones achieved for each child is modelled through the sequential updating of the prior.”

a. It’s unclear that the prior will not overwhelm the likelihood if subjects fail the first t trials, succeed at t+1.

b. If I understand correctly, the milestones are tested monthly, with 3 milestones tested each month for the first 12 months. Is it reasonable to assume these are ordered appropriately and that the independence assumption holds? Was a sensitivity analysis performed to assess the impact of this assumption? If not, it would be suggested to do so.

2. When did most children start the program? How much data is missing from the beginning of the trajectories and does this correlate with the groupings? Can the authors elaborate on this in the paper?

3. Dahl (2006) has a method that uses the similarity matrix as well. It selects the observed clustering that minimizes the sum of squared deviations from the similarity matrix; therefore, the method requires less user input and avoids the need to select the total number of clusters, k. Perhaps, the authors should explore compare the PAM results to this method, which should be fairly easy since they have the similarity matrix already.

4. In the grid experiment for the hyperparameters, it appears that only informative priors are suitable. How did the model perform under noninformative priors?

5. The ESS relative to the number of samples seems fairly low. Did you consider alternative sampling schemes to increase the mixing?

6. What are the posterior distributions for the group means for the resulting clusters? Are all samples used, only those that coincide with the number of samples k or with the same arrangement, or some other means? This should be clarified in the text.

Reviewer #2: My one minor comment is that it's a bit confusing when you list the hyperparameters (page 13) used and the last one is "\\alpha=Gamma(1,1)". I think it would be better to say "\\eta_1=\\eta_2=1" or "\\alpha has a Gamma(1,1) distribution".

6. PLOS authors have the option to publish the peer review history of their article (what does this mean?). If published, this will include your full peer review and any attached files.

Reviewer #1: Yes: Amy LaLonde, PhD

Reviewer #2: No

---

## [Author Response · Author response to Decision Letter 0]

23 Feb 2020

Response to reviewers

Manuscript PONE-D-19-29401

Identifying latent subgroups of children with developmental delay using Bayesian sequential updating and Dirichlet process mixture modelling

A copy of our revised manuscript, including revised supplementary material, has been provided in the document labelled “Manuscript”. In addition, a copy of the manuscript with changes highlighted is also provided in the document labelled “Revised manuscript with track changes”. In order to not increase the overall length of the manuscript, we have added additional supplementary material or have placed some results of analyses on the first author’s Github. We have indicated where we have implemented the changes in each response to the reviewer’s comments; line numbers are those corresponding to the “Manuscript” document.

We would like to thank the Academic Editor, Man Ki Kwok; Reviewer 1, Amy LaLonde; and Reviewer 2, for providing the detailed comments to improve our manuscript. We feel that our revisions based on these comments and suggestions has lifted the quality of our paper. Following are our systematic responses to the comments. Reviewers’ comments are italicised, with our response given immediately below the italicised text.

Response to Academic Editor Comments

1a. Timing of milestones:

This study developed a Bayesian sequential updating modeling which counted the cumulative developmental milestone achievement over the first 3 years of life in 79 young children with different levels of development and disability in Australia. However, timing was not considered and hence earlier detection of deviation from typical development could not be achieved for early intervention.

Thank you for your comment on timing of the milestones. In order to develop the method we used a static data set provided by the Developing Foundation and have used the complete data sequence for each child to calculate the area, which we view as a summary of the development of that child up to the time point of interest. We agree that with a retrospective design, such as this, it would not be possible to identify the delay at an earlier age (particularly when using the full three years of data). However, the intention would be for the Developing Foundation to use this method prospectively, to be able to monitor the development of children in real time, using the sequential updating. 

In order to emphasise this distinction, we have added the following to the Discussion (Line 401):

“To develop the method, a static data set was used and the method was implemented retrospectively, however, by using Bayesian sequential updating, this method could be implemented prospectively, where predictions could be made as the child develops, as the method allows for predictions to be easily updated with the collection of new data.”

We hope this sufficiently addresses your concerns and please clarify if we have not interpreted this comment correctly.

1b. Alternative Modeling:

The authors should consider assessing the modeling performance by comparing conventional standard methods for generating latent classes with the studied Bayesian sequential updating modeling for adding subsequent milestones and the Dirichlet process mixture modeling for clustering deviated area from typical development.

Thank you for this suggestion. We have implemented a model comparison where we compared the clusters that we obtained from the DPMM to K-means and model-based clustering. An additional Appendix (S3_Appendix) has been added to the supplementary materials which contains a detailed explanation and analysis of the clustering using these two alternative clustering algorithms. 

In addition, we have briefly summarised the results of this analysis in the manuscript by adding (Line 281):

“Additional analyses using two alternative clustering algorithms, namely, k-means and model-based clustering, were undertaken to compare the performance of the DPMM to these commonly used clustering methods. In summary, K-means was only able to cluster the data into two groups, representing typical and atypical development. Model-based clustering performed slightly better as it was able to identify two smaller clusters, in addition to the typical/atypical clusters. These smaller clusters were equivalent to clusters 5 and 8 from the DPMM. Overall, the DPMM was able to identify smaller, more distinct clusters than these alternative methods, which is important for the current application where identifying smaller groups for targeted intervention is the goal. A detailed overview of these methods and the results of these additional analyses can be found in S3 Appendix.”

Response to Reviewer 1 Comments

1. Is the assumption of independence between the same milestone measurements over time reasonable? The author’s state “This is considered a reasonable assumption as milestone achievement is not necessarily cumulative, in that some children can achieve later milestones without achieving earlier ones. Moreover, the dependency between milestones achieved for each child is modelled through the sequential updating of the prior.”

1a. It’s unclear that the prior will not overwhelm the likelihood if subjects fail the first t trials, succeed at t+1.

Thank you for your comment, we have simulated the example outlined in this comment, to demonstrate the behaviour of the prior. The figure below is a simulated example where a child has failed the first 9 milestones and succeeded at milestone 10 and 11. The initial prior follows a Beta(1,1) distribution which is an uninformative uniform prior, with a mean of 0.5. We can see that with each subsequent failure the probability of achieving the next milestone decreases and the range of uncertainty also decreases (as indicated by the 95% highest density region). However, at time points 10 and 11, the probability of achieving the next milestone begins to increase and the uncertainty also increases. The increase is not very large, since the child has failed in 9 previous trials. This, to us, is one of the main advantages of sequential updating as the prior incorporates the information from previous trials so each new measurement depends on the history for each child. 

We chose the initial prior to be a Beta(1,1) distribution, which is a uniform prior. This translates to a 50% probability of achieving the first milestone, with the largest possible uncertainty. We wanted to be as conservative as possible with this initial prior, to not influence the sequences in either direction before seeing any data. This prior could be changed in the future to an informative prior, perhaps reflecting the probability of milestone achievement in the target population. 

We hope this explanation has satisfied the reviewer’s concerns regarding the prior used for the sequential updating. We have not modified anything in the manuscript as we believe our explanation of the sequential updating of the prior and the examples in Figure 1 are enough to demonstrate this process. We could, however, add the above exposition to the Supplementary Material if the reviewer and editor believe that it is useful. Please inform us if so, or if further clarification is needed.

1b. If I understand correctly, the milestones are tested monthly, with 3 milestones tested each month for the first 12 months. Is it reasonable to assume these are ordered appropriately and that the independence assumption holds? Was a sensitivity analysis performed to assess the impact of this assumption? If not, it would be suggested to do so.

Thank you for the suggestion. The milestones were ordered by developmental experts at the Developing Foundation. In order to reassure the reader that the independence assumption is reasonable, we have performed a sensitivity analysis, as suggested. We randomised the order of milestones within each month, performed the sequential updating and recalculated the areas. We then compared the difference between the original areas and the areas obtained from the random ordering. We then repeated this process 10 times with a new random ordering. The results showed negligible differences in the areas across all functional domains and random orders.

We have added the following lines to the methods section of the manuscript (Line 135):

“However, in order to investigate the independence assumption, a sensitivity analysis was performed, and the results indicated that this assumption is reasonable. This analysis is addressed in more detail in the Sensitivity analyses section of the Results.”

We also added the following to the Sensitivity analyses section of the Results (Line 301):

“Two sensitivity analyses were performed to assess different aspects of the model. In the first sensitivity analysis, two simulation studies were performed to illustrate how the proposed DPMM performs for (a) well separated, adjacent or overlapping clusters (scenario 1) and (b) small, medium or large sample sizes (scenario 2). The second sensitivity analysis assessed the independence assumption of the milestones and is briefly described at the end of this section.”

And (Line 377):

“A second sensitivity analysis was performed to assess the independence assumption of the milestone measurements. The order of the milestones within each month was randomised in order to investigate the impact of rearranging the order of the milestones. The sequential updating and area calculation were performed on the rearranged milestones, and the results compared to the original ordering of the milestones. This procedure was repeated 10 times with new random orders. The results revealed only minor differences between the areas obtained from the original order and the random order (overall mean difference = -0.00047, overall mean standard error =0.00093). The full results of this sensitivity analysis can be found on the first author's Github.”

We also added to the Discussion (Line 420):

“Firstly, the developmental milestones within each month were assumed to be sequential, based on information elicited from a domain expert. However, the milestones may not be met in this exact order for every child. A sensitivity analysis assessing the independence assumption revealed only small differences in the outcome if the milestones were rearranged within each month. Given this small difference, it is reasonable to assume independence for the milestones within each month and a more general model could use the binomial distribution to model the milestones, but this approach was not considered here”

We have also added the following to the description of the milestones (Line 109):

“The order of the milestones was determined by developmental experts at The Developing Foundation.”

2. When did most children start the program? How much data is missing from the beginning of the trajectories and does this correlate with the groupings? Can the authors elaborate on this in the paper?

Thank you for pointing this out. We realise our explanation in the paper was a little vague. The 79 children in this study have all started at month 1, so there is no data missing from the beginning of the trajectories. If there were any children who had started later, we planned to account for this by moving the starting point of the reference sequence to the same month as the child’s sequence. We have updated our explanation of this in the Methods which now reads (Line 176):

“In this application, all 79 children started their milestone measurements in month 1, but this may not always be the case. If children do not begin their milestone measurements in the first month (i.e., there are measurements missing before the beginning of the sequence), the starting point for the reference sequence can be set equal to the starting point of the child's sequence, in order for the reference sequence to remain the same for all children.”

3. Dahl (2006) has a method that uses the similarity matrix as well. It selects the observed clustering that minimizes the sum of squared deviations from the similarity matrix; therefore, the method requires less user input and avoids the need to select the total number of clusters, k. Perhaps, the authors should explore compare the PAM results to this method, which should be fairly easy since they have the similarity matrix already.

Thank you for the suggestion. We implemented the method by Dahl (2006) as suggested but found that it returned many more clusters than PAM. When processing each chain individually the method returned 14, 13, and 16 clusters, respectively, and when processing the average posterior similarity matrix from the combined chains the method returned 19 and 20 clusters. Many of these clusters were of size N=1 and the number of clusters was considered excessive for the current application. Therefore, we have briefly described this comparison in the manuscript, but have not gone into too much detail. We have added (Line 292):

“In addition, we compared the PAM post-processing method to the least-squares clustering approach (Dahl, 2006). Briefly, this method selects a clustering which minimizes the sum of squared deviations from the posterior similarity matrix. This method returned many more clusters than the PAM method, with an average of 16 clusters selected. This was the case when examining each chain separately, as well as combined. Sixteen clusters were considered excessive for the current application, and many of these clusters had a sample size of N=1. Therefore, the clusters obtained from this method were not explored further.”

4. In the grid experiment for the hyperparameters, it appears that only informative priors are suitable. How did the model perform under noninformative priors?

The hyperparameters that were selected for the grid experiment were chosen based on examples found in the literature. We tried to vary some parameters as much as possible in order to observe their influence on the model. To our knowledge, noninformative priors are not often used in DPMM. Many studies do not state how they chose the hyperparameters or they state that they selected them through trial and error. We have taken the latter approach but attempted to do this more rigorously through the grid experiment. There is one noninformative prior for the inverse-Wishart distribution for the prior on the variance-covariance matrix which we did not initially include. This is the IW(d+1,I), with d+1 degrees of freedom, where d is the number of dimensions of the data. We have performed a smaller grid experiment using this prior, where we varied the value for N0 as either 1, 0.1 or 0.01 and we specified the prior for α as either Gamma(1,1) or Gamma(2,2) as per the original grid experiment. This resulted in 6 models in total. The two models with an N0 parameter of 0.1 did not converge, due to different chains exploring two separate modes for the number of clusters. The remaining four models returned three clusters of size 68, 9 and 3, which are not sensible clusters. The noninformative inverse-Wishart prior only appears to identify extreme groups while placing the remaining participants into one large cluster. As this cluster configuration has very limited practical application, we have decided not to include these additional models to the grid experiment. 

We have added additional references to justify our choice of hyperparameters by adding (Line 237):

“The selection of hyperparameters chosen for the grid experiment were guided by the literature, where similar hyperparameters have been used [48, 52-56].”

We have not included these additional models to the paper as they did not return sensible clusters, so the results do not add to the overall message of the paper. We also decided not to add this to the Supplementary Material as we did not want to add to the already quite lengthy analysis and discussion. Please let us know if you would prefer us to add these additional models to the S2_Appendix.

5. The ESS relative to the number of samples seems fairly low. Did you consider alternative sampling schemes to increase the mixing?

We did consider alternative sampling schemes, however, we decided that we would likely encounter similar mixing difficulties regardless of which sampler was used. This issue is discussed in Hastie et al (2015), where they state that “For real data applications of the DPMM, the state space can be highly multimodal with well separated regions of high posterior probability coexisting, often corresponding to clusterings with different number of components (p. 1024)”. They also conclude that DPMM samplers perform well on simulated datasets but do not always perform as well when applied to real-data examples (Hastie et al., 2015). Rather than use an alternative sampler, which would most likely not improve the mixing, we followed the advice of Hastie et al. (2015), by running multiple chains, each initialised at a different number of clusters, implemented label switching moves, and extended the number of iterations from 100,000 to 1,000,000. We have also acknowledged the limitations of the slice sampler and have addressed these limitations in the Discussion as follows (Line 448):

“In this paper, the slice sampler was selected as the method for sampling from the posterior distribution of the Dirichlet process. There are, however, several alternative samplers that can be used, for example, the truncated sampler [43] and the retrospective sampler [44]. The slice sampler was used in this application as it adaptively selects the number of mixture components [67] and easily updates them at each iteration [68]. Also, unlike the truncated methods, it targets the true posterior rather than an approximation [69]. However, this method does have some limitations. Due to the high correlation between each slice from the slice sampler and the mixture weights, the number of components sampled at each iteration can be large if the slice is small [69, 70]. This can result in slow mixing and high autocorrelations [67], as was the case in this research. However, as these samplers are often developed and illustrated on simulated or low-dimensional datasets, it is likely that similar problems would be encountered using alternative samplers when applied to complex data, such as that used in the current application [45].”

As we feel that we had already addressed the limitations of the slice sampler in the Discussion as above, we have not added any additional explanation. Please let us know if you would like further elaboration. 

Reference:

Hastie, D. I., Liverani, S., & Richardson, S. (2015). Sampling from Dirichlet process mixture models with unknown concentration parameter: mixing issues in large data implementations. Statistics and computing, 25(5), 1023-1037.

6. What are the posterior distributions for the group means for the resulting clusters? Are all samples used, only those that coincide with the number of samples k or with the same arrangement, or some other means? This should be clarified in the text.

It is not possible, using DPMM, to obtain posterior distributions for the group means. This is because the number of groups and the composition of the groups change at every iteration. In addition, we have implemented label switching moves in order to improve the mixing of the sampler. This means that the cluster labels also may change at each iteration, so we cannot use the samples that correspond to the number of clusters, k, to calculate these posterior distributions. 

The means and standard deviations reported in Table 2 are the sample mean and standard deviation for each group. We have changed the wording in the text to make this more explicit by stating (Line 254): “The sample means and standard deviations for the areas of each group, as well as the group sizes can be found in Table 2”.

Response to Reviewer 2 Comments

1. My one minor comment is that it's a bit confusing when you list the hyperparameters (page 13) used and the last one is "\\alpha=Gamma(1,1)". I think it would be better to say "\\eta_1=\\eta_2=1" or "\\alpha has a Gamma(1,1) distribution".

As per the above suggestion, we have changed this line in the manuscript which now reads (Line 241): “In addition, the prior distribution for α was a Gamma(1,1) distribution.”

Response to Journal requirement comments

We have done our best to follow the style and file naming requirements for PLOS ONE. We have not been able to identify any deviations from the style requirements. Please let us know if there are specific parts of the manuscript which are not meeting these requirements.

2. You indicated that you had ethical approval for your study.

In your Methods section, please ensure you have also stated whether you obtained consent from parents or guardians of the minors included in the study or whether the research ethics committee or IRB specifically waived the need for their consent.

We have added to the methods section (Line 121): “The QUT University Human Research Ethics committee waived the need for consent from the parents or guardians for the data used in this research, as the data does not contain any identifiable information.”

'Work by Patricia Gilholm was supported by an Australian Technology Network of Universities Industry Doctoral Training Centre scholarship, co-funded by QUT and the Developing Foundation. Furthermore, the data used in this research was generously provided by The Developing Foundation.'

"Work by PG is partially funded by the Developing Foundation (https://www.developingfoundation.org.au/). The Developing Foundation also generously provided the data for this study."

We have removed the funding-related text from the manuscript which now reads (Line 494) “The data used in this research was generously provided by The Developing Foundation”.

3a. Please provide an amended Funding Statement that declares *all* the funding or sources of support received during this specific study (whether external or internal to your organization) as detailed online in our guide for authors at http://journals.plos.org/plosone/s/submit-now

3b. Please state what role the funders took in the study. If any authors received a salary from any of your funders, please state which authors and which funder. If the funders had no role, please state: "The funders had no role in study design, data collection and analysis, decision to publish, or preparation of the manuscript."

3c. Please include your amended statements within your cover letter; we will change the online submission form on your behalf.

Please refer to our amended statement below:

“Work by PG was supported by an Australian Technology Network of Universities Industry Doctoral Training Centre scholarship, co-funded by QUT and the Developing Foundation. The Developing Foundation played a role in data collection. The funders had no role in study design, analysis, decision to publish or the preparation of the manuscript”. 

 In your revised cover letter, please address the following prompts

4a. If there are ethical or legal restrictions on sharing a de-identified data set, please explain them in detail (e.g., data contain potentially sensitive information, data are owned by a third-party organization, etc.) and who has imposed them (e.g., an ethics committee). Please also provide contact information for a data access committee, ethics committee, or other institutional body to which data requests may be sent.

4b. If there are no restrictions, please upload the minimal anonymized data set necessary to replicate your study findings as either Supporting Information files or to a stable, public repository and provide us with the relevant URLs, DOIs, or accession numbers. For a list of acceptable repositories, please see http://journals.plos.org/plosone/s/data-availability#loc-recommended-repositories.

Please see our amended Data Availability statement below. We hope this amended statement meets the requirements. Please let us know if there is anything further that we need to add to this statement:

“The data used in this study contains milestone measurements of infants and young children aged between 1 month and 3 years. This data has been provided and is owned by The Developing Foundation. Therefore, the authors of this study cannot legally distribute this data. To gain access to the data, interested researchers can contact the Data and Research Manager at The Developing Foundation, Hugh McKenzie (hugh@developingfoundation.org.au).”

---

## [Decision Letter · Decision Letter 1]

23 Mar 2020

PONE-D-19-29401R1

Identifying latent subgroups of children with developmental delay using Bayesian sequential updating and Dirichlet process mixture modelling

PLOS ONE

Dear Miss Gilholm,

Thank you for submitting your manuscript to PLOS ONE. After careful consideration, we feel that it has merit but does not fully meet PLOS ONE’s publication criteria as it currently stands. Therefore, we invite you to submit a revised version of the manuscript that addresses the points raised during the review process.

For transparency and replication, the authors are encouraged to prepare R codes (with sample data) for the three method sections: the Bayesian sequential updating, area between posterior probability sequences, and Dirichlet process mixture model and make them publicly available. Currently, only the slice sampling code is publicly available on Github.

We would appreciate receiving your revised manuscript by May 07 2020 11:59PM. To enhance the reproducibility of your results, we recommend that if applicable you deposit your laboratory protocols in protocols.io, where a protocol can be assigned its own identifier (DOI) such that it can be cited independently in the future. For instructions see: http://journals.plos.org/plosone/s/submission-guidelines#loc-laboratory-protocols

We look forward to receiving your revised manuscript.

Kind regards,

Man Ki Kwok

Academic Editor

PLOS ONE

Journal Requirements:

Additional Editor Comments (if provided):

Reviewers' comments:

Reviewer's Responses to Questions

**Comments to the Author**

1. If the authors have adequately addressed your comments raised in a previous round of review and you feel that this manuscript is now acceptable for publication, you may indicate that here to bypass the “Comments to the Author” section, enter your conflict of interest statement in the “Confidential to Editor” section, and submit your "Accept" recommendation.

Reviewer #2: All comments have been addressed

2. Is the manuscript technically sound, and do the data support the conclusions?

Reviewer #2: Yes

3. Has the statistical analysis been performed appropriately and rigorously? 

Reviewer #2: Yes

4. Have the authors made all data underlying the findings in their manuscript fully available?

Reviewer #2: Yes

5. Is the manuscript presented in an intelligible fashion and written in standard English?

Reviewer #2: Yes

6. Review Comments to the Author

Reviewer #2: (No Response)

7. PLOS authors have the option to publish the peer review history of their article (what does this mean?). If published, this will include your full peer review and any attached files.

Reviewer #2: No

---

## [Author Response · Author response to Decision Letter 1]

17 Apr 2020

Response to Academic Editor Comments

1. For transparency and replication, the authors are encouraged to prepare R codes (with sample data) for the three method sections: the Bayesian sequential updating, area between posterior probability sequences, and Dirichlet process mixture model and make them publicly available. Currently, only the slice sampling code is publicly available on Github.

Thank you for this suggestion. We had R code for performing the Bayesian sequential updating and area calculation on the first author’s Github, but this was not referred to explicitly in the manuscript. In addition, we have prepared an R markdown walk through, using simulated data, of the analyses performed in the paper. We have added the following line to the manuscript directing the reader to the Github account, where they can find this information (Line 235):

“R code for performing the Bayesian sequential updating and area calculation, as well as an example of using this code on simulated data is available on Github.”

---

## [Editor Report · Decision Letter 2]

8 May 2020

Identifying latent subgroups of children with developmental delay using Bayesian sequential updating and Dirichlet process mixture modelling

PONE-D-19-29401R2

Dear Dr. Gilholm,

We are pleased to inform you that your manuscript has been judged scientifically suitable for publication and will be formally accepted for publication once it complies with all outstanding technical requirements.

With kind regards,

Man Ki Kwok

Academic Editor

PLOS ONE
---

## [Editor Report · Acceptance letter]

20 May 2020

PONE-D-19-29401R2 

Identifying latent subgroups of children with developmental delay using Bayesian sequential updating and Dirichlet process mixture modelling. 

Dear Dr. Gilholm:

I am pleased to inform you that your manuscript has been deemed suitable for publication in PLOS ONE. Congratulations! Your manuscript is now with our production department. 

With kind regards,

on behalf of

Dr. Man Ki Kwok 

Academic Editor

PLOS ONE